# Use of a Local Anaesthetic/Antiseptic Formulation for the Treatment of Lambs Experimentally Infected with Orf Virus

**DOI:** 10.3390/ani13182962

**Published:** 2023-09-19

**Authors:** Delia Lacasta, Marina Ríos, Marta Ruiz de Arcaute, Aurora Ortín, Juan José Ramos, Sergio Villanueva-Saz, María Teresa Tejedor, Héctor Ruiz, Marta Borobia, Ramsés Reina, Alex Gómez, Teresa Navarro, Peter Andrew Windsor

**Affiliations:** 1Animal Pathology Department, Instituto Agroalimentario de Aragón-IA2 (Universidad de Zaragoza-CITA), Veterinary Faculty of Zaragoza, C/Miguel Servet 177, 50013 Zaragoza, Spain; marinarl24199@gmail.com (M.R.); martarda@unizar.es (M.R.d.A.); aortin@unizar.es (A.O.); jjramos@unizar.es (J.J.R.); svs@unizar.es (S.V.-S.); hectorruiz353@gmail.com (H.R.); mborobia@unizar.es (M.B.); alex.gmezcalvo@gmail.com (A.G.); teresanarr@gmail.com (T.N.); 2Anatomy, Embryology and Animal Genetics Department, CIBER CV (Universidad de Zaragoza-IIS), Veterinary Faculty of Zaragoza, C/Miguel Servet 177, 50013 Zaragoza, Spain; ttejedor@unizar.es; 3Instituto de Agrobiotecnología, CSIC-Gobierno de Navarra, 31192 Mutilva, Spain; ramses.reina@csic.es; 4Sydney School of Veterinary Science, The University of Sydney, Camden, NSW 2570, Australia; peter.windsor@sydney.edu.au

**Keywords:** lambs, contagious ecthyma, local anaesthetic/antiseptic treatment, experimental infection

## Abstract

**Simple Summary:**

Contagious ecthyma (orf) is a highly contagious eruptive skin infection of sheep and goats caused by the orf virus of the genus *Parapoxvirus. *Orf mainly affects young animals in their first year of life, although infection has occurred in other animals and in humans, so it is considered a zoonosis. The disease has high economic repercussions on sheep and goat farms worldwide and can compromise small ruminant value chains, so effective control and therapy are required. As treatment of orf lesions with topical antibiotics is a common practice, non-antimicrobial alternatives are indicated to promote antimicrobial stewardship and avoid increased antimicrobial resistance. A study with the wound formulation Tri-Solfen^®^, a topically applied product containing two local anaesthetics (lignocaine and bupivacaine), adrenaline and an antiseptic (cetrimide) in a gel matrix, was conducted. This product was developed for surgical husbandry procedures in Australian livestock and more recently shown to be efficacious for the treatment of foot-and-mouth disease (FMD) in large ruminants, providing almost instant pain relief and faster healing of wounds with a likely viricidal impact. Results of therapy in lambs experimentally infected with orf appeared less encouraging, likely reflecting the relatively milder clinical signs than bovine FMD and deep basal epithelial location of orf lesions.

**Abstract:**

Contagious ecthyma is a highly transmissible eruptive viral disease of the skin and mucosa of sheep and goats distributed worldwide. The treatment of orf lesions is usually based on the use of antiseptics and antibiotics for the management of presumptive secondary infections, increasing risks of antimicrobial resistance. The wound dressing formulation Tri-Solfen^®^ (TS) containing two local anaesthetics (lignocaine and bupivacaine), adrenaline and an antiseptic (cetrimide) in a gel formulation has been demonstrated to reduce suffering and enhance recovery in cattle and buffalo with oral and skin lesions due to foot-and-mouth disease virus infection and reduced the orf viral load in lambs. In the present study, experimental infection with the orf virus was conducted in 50 newborn lambs and 25 animals were treated after the presence of the first lesions with TS and repeated three days later. Daily clinical examination, haematological, serological, biomolecular and post-mortem analyses were conducted during 34 days after treatment. Results indicated that treatment had no effect on weight gain and clinical progression of the lesions. It was determined that seroconversion after experimental infection occurs 34 days after infection and suggested that the deep basal epithelial location of the orf lesions may have prevented the therapy from having altered the clinical course.

## 1. Introduction

Contagious ecthyma (also known as orf) is a highly contagious zoonotic disease of sheep and goats that affects mainly young animals in the first year of their life, caused by the Orf virus (ORFV) belonging to family *Poxviridae*, sub-family *Chordopoxvirinae*, and genus *Parapoxvirus* [1]. Orf has a worldwide distribution. It causes significant financial losses in livestock production, particularly in countries without routine vaccination and where severe outbreaks are generally associated with intensive sheep husbandry. Orf is included in the list of diseases with the most significant economic repercussions in sheep and goat farms, causing direct losses due to high mortality [2] and impacts on trade [3]. It is also associated with indirect losses from weight loss and a predisposition to suffer from other diseases due to immunosuppression, compromising production and profit margins from increasing costs associated with food, medicines, and veterinary services [4]. Further, orf is recognised as a common zoonosis. It causes painful skin lesions in people in contact with ORFV from infected animals or self-administration when using live orf vaccine, especially farmers and veterinarians. Lesions in humans are similar to those observed in infected animals, although generally milder, appearing 3 to 7 days after infection and persisting for weeks, most often located at contact sites on the hands and arms. Sometimes these lesions are described as markedly granulomatous and can take months to heal [1].

ORFV is epitheliotropic and replicates into the epithelial cells and keratinocytes, evading the defensive barrier of the host [5]. The infected epidermis is characterised by vacuolisation and swelling of keratinocytes. Infectious virions are detected 12 h after infection, and the maximum titre has been reported to be between 24–72 h post-infection [6]. Dendritic cells accumulated at the site of infections do not have access to T-cells in the lymph nodes, with infection generally localised [7]. Loss of epithelial integrity is the main predisposing factor in the initiation of the disease, along with stress factors, including transportation, immune suppression or other primary infections [1].

Clinical presentation in lambs or kids includes the formation of vesicles, papules, pustules and scabs, sometimes proliferative, on the skin and mucous membranes of affected animals. Although the most frequent location of these lesions is in the mouth, they can also be found on the skin of the limbs, perianal area, scrotum, ears, eyes, etc. In adult animals, mammary skin location in sheep is very common [8]. The painful lesions that affect the mammary gland cause the ewe to prevent the lamb from nursing normally. The hungry lambs or kids try to find milk in other dams, contributing to the spread of the disease. The clinical disease usually lasts 3–4 weeks with lesions resolving in 1–2 months [1].

Transmission of orf is from direct or indirect contact with the orf virus from cutaneous lesions of infected animals containing the virus, or live virus during vaccination. Orf virus is very resistant in the environment, mainly in dry conditions, surviving for months and even years, although persistence may be shorter in cold and wet conditions [8]. Persistently infected carrier sheep are probably responsible for the reservoir and reappearance of contagious ecthyma within flocks [9].

Clinical diagnosis of contagious ecthyma is straightforward. However, laboratory confirmation is essential as some differential diagnoses are notifiable diseases, especially foot-and-mouth, sheeppox, bluetongue, and peste des petits ruminants. Current laboratory diagnostics include PCR, ELISA, electron microscopy, histopathology, and Western blotting [10]. Nevertheless, molecular methods are considered the most helpful tool for detecting this virus at the herd level. Viral DNA amplification and detection techniques, particularly polymerase chain reaction (PCR) tests based on the detection of the B2L gene, are the most widely used. The advantage of this technique is that it is a direct method capable of detecting viral DNA in animal tissues and fluids before seroconversion [11].

Although vaccination is the preferred option to control the disease, the orf vaccine is currently unavailable in most European countries. Where available, existing live-attenuated or scab-based vaccines are used to effectively suppress the disease, despite not invoking full protection [12]. Importantly, there is no effective treatment against the contagious ecthyma virus. Symptomatic treatment with dressings and local antiseptics has been used since it is generally a self-limiting disease, although topical and systemic antibiotics have been repeatedly recommended and promoted as reducing secondary bacterial contamination in ORFV infections. With the emergence of antimicrobial stewardship in response to the priority one health issue of managing antimicrobial resistance risk, there is a need for new treatment protocols that avoid the use of antibiotics. 

Several studies have confirmed the efficacy of the wound therapy formulation Tri-Solfen^®^ (TS) (Animal Ethics Pty Ltd., Sydney, Australia) for reducing pain and hastening the healing of skin and mucosal lesions in sheep and cattle [13,14,15]. This formulation contains two local anaesthetics (lignocaine and bupivacaine), adrenaline and an antiseptic (cetrimide) in a gel formulation that creates a barrier effect, blocking nociception and numbing the pain of lesions. TS was developed for pain relief in sheep undergoing surgical husbandry procedures in Australia. However, it has been shown to be efficacious in other applications, including foot and mouth disease (FMD) therapy in order to reduce suffering and enhancing recovery [13,16]. Due to its low pH (2.7), it has a potential viricidal effect that could reduce the risk of transmission, avoiding the need of using antibiotics.

In the present study, an experimental infection with the wild-isolated ORFV was conducted on 50 lambs with the aim of making batches as homogeneous as possible to test the efficacy of this topical treatment.

## 2. Materials and Methods

In the present study, experimental infection with the orf virus was conducted in 50 newborn lambs from a commercial sheep farm without orf outbreaks during the last three years. Once the first orf lesion appeared, half of the lambs (25) were treated with TS^®^ and the remaining 25 were kept as the control group. In both batches, the evolution of the lesions and the presence and viability of the virus were analysed for one month until their humanitarian sacrifice.

All the procedures were supervised and approved by the Ethics Advisory Commission for Animal Experimentation (nº PI33/21), the Biosafety Committee and the Occupational Risk Prevention Unit of the University of Zaragoza, in accordance with current regulations regarding these procedures. aspects (R.D. 53/2013, Law 31/1995, R.D. 664/1997, R.D. 1299/2006).

### 2.1. Studied Lambs and Weighing

After a correct colostrum intake, 50 healthy newborn lambs were recruited from a Lacaune dairy sheep farm. The farm had not had orf outbreaks, at least during the last three years. However, serological and molecular analyses were also conducted on the lambs to ensure the absence of the virus and the antibodies against the virus. 

The selected lambs were four-day-old males with an average weight of 5.85 kg on arrival at our facilities. All the animals were PCR and ELISA negatives before being transferred to the facilities of the Veterinary Faculty, where they were registered with individual ear tags for easy identification and randomly distributed in two independent and completely isolated boxes, with 25 lambs in each (Box 1 and Box 2) for the entire study duration. The weights of all the lambs were measured on the day of their reception and weekly throughout the study period.

Forty lambs (20 of each group) were euthanised one month after the experimental infection. However, the remaining 10 lambs (5 per group) were retained for an additional month to follow the evolution of antibodies against the orf virus and finally also euthanised.

### 2.2. Experimental Infection

The ORFV used for the experimental infection was isolated by standard methods [17] from scab specimens collected from skin lesions of *Rasa Aragonesa* lambs affected by a natural CE (contagious ecthyma) outbreak in Navarra (Spain). Infection was confirmed and characterised by PCR using primers for ORFV 045 gene [11]. Subsequently, sequencing analysis was performed. Field virulent ORFV was propagated on KOP-R (bovine oesophagus) cells, and cell culture supernatant from infected cells was collected when approximately 90% of the culture showed cytopathic effect (CPE) and stored at −80 °C. For titration, KOP-R cells were trypsinised and passaged to 96-well plates (0.16 cm ^2^/well) at a density of 1.5 × 10^6^ cells/cm^2^ and incubated under 5% CO_2_ at 37 °C conditions. Subsequently, KOP-R cells were infected with serial 10-fold dilutions (10^−1^ to 10^−12^) of ORFV and maintained at 37 °C with 5% CO_2_ with Dulbecco’s modified Eagle’s medium (DMEM) supplemented with 1% heat-inactivated FBS (fetal bovine serum), 0.1% L-glutamine, and 0.1% antibiotics/antimycotics mix (Sigma Aldrich, St. Louis, MO, USA) (1% DMEM). CPE was tested after 7 days and the Reed–Müench method [18] was used to calculate the TCID_50_ (Median tissue culture infectious dose), reaching 10^6^ TCID_50_/mL. After titration, collected supernatants were diluted in DMEM and distributed in aliquots of 1 mL containing 10^4^ TCID_50_/mL of ORFV.

As previous work conducted by our team to determine the best route of virus infection identified that the intradermal injection offered the most efficient experimental infection for reproducing standard CE lesions (unpublished data), experimental infection with ORFV involved injecting 1 mL of inoculum with an intradermal inoculation gun. This device enables a very fine, pressurised stream to perforate the skin and produce a characteristic papule containing the inoculum. Aliquots of 1 mL containing 10^4^ TCID_50_/mL of ORFV were used for the inoculation. The animals were manually restrained during the procedure to ensure correct immobilisation during inoculation. An average of 17 inoculations were administered for each lamb (1.7 mL/animal), split between the right, left and central upper lips; right, left and central lower lips; and the upper and lower aspects of the interior of the mouth (gums and/or palate). 

### 2.3. Treatment Application

Once vesicles of contagious ecthyma were detected 8 days after the experimental infection, and orf infection was confirmed by PCR, the animals from Box 2 were treated by spraying 1.5mL of TS^®^ (Sydney, Australia) using a dosing gun and spreading the product so that all lesions on the lips and inside of the mouth were covered with the product. The treatment was repeated three days later (11 days post-inoculation) in the affected areas of the face. The 25 lambs from Box 1 were kept as a control group, and no treatment was applied.

### 2.4. Clinical Examination

The lambs were examined daily for the presence of orf lesions. In addition, photos were taken daily for further detailed study. In each lamb, both profiles were photographed—the front part of the mouth and the area of the gums and teeth. Subsequently, the photos were grouped by lamb based on the numbering of the ear tags. For their subsequent statistical study, the images were analysed individually, and the lesions were numerically coded based on the areas in which they appeared and their degree of severity. In this way, they were classified into right upper lip (RUL), left upper lip (LUL), right lower lip (RLL), left lower lip (LLL), gums, and “other locations” such as the nose, eyes, face, or prepuce. Likewise, they were classified according to the type of lesion (a: “papules or pustules”, b: “scabs”, or c: “absence of lesions”) and its severity (from 0 to 5, assuming 0 “absence of injuries”, 1 “mild injuries” and 5 “very serious injuries”). In summary, an individual daily record was obtained for each lamb showing the location, severity, and progression of the lesions throughout the entire study.

### 2.5. Haematological Analysis

Whole blood samples were collected from the jugular vein into EDTA anticoagulant tubes from all studied animals for subsequent haematological analysis. Samples were collected weekly, once prior to experimental infection (He0), again a week later, when the lesions began to appear but prior to treatment (He1), and twice after treatment (He2, 7 days post-treatment and He3, 13 days post-treatment).

Haematology was performed with an IDEXX ProcyteDx automatic haematology counter (IDEXX laboratories, Westbrook, ME, USA). Measured parameters included leukocytes (K/mL), erythrocytes (M/μL), haemoglobin (g/dL), haematocrit (%), platelets (K/μL), VCM (Mean Corpuscular Volume; fL), HCM (Corpuscular Hemoglobin Mean; pg), MCHC (Mean Corpuscular Hemoglobin Concentration; g/dL) and reticulocytes (K/μL). White series blood cells were also evaluated by counting neutrophils (K/μL), lymphocytes (K/μL), monocytes (K/μL), basophils (K/μL), and eosinophils (K/μL).

### 2.6. Viral Quantification by PCR

In order to detect the presence of the virus in the infected skin and mucous membranes, samples were collected using sterile swabs preserved in a culture medium (Deltalab). The swabs were rotated on the infected skin before the lesions appeared and on several of the lesions observed in each lamb after the lesions appeared. Samples were obtained from all animals before infection and treatment (H0) and after application of TS^®^: H1 (15 days post-infection), H2 (21 days post-infection), and H3 (35 days post-infection). The H3 sample was taken only from the 10 lambs that were still alive one month after infection to continue with their serological study.

Quantitative PCR was based on nucleic acid extraction using the MagMAX™ Pathogen RNA/DNA commercial kit (Thermo Fisher Scientific, Vantaa, Finland), and the KingFisher Flex System automated magnetic particle processor (Thermo Fisher Scientific), following manufacturer’s instructions. The extracted DNA samples were stored at −80 °C until the end of the sampling to be able to evaluate all in a single real-time PCR run. 

The orf virus was detected using the commercial qPCR kit EXOone Contagious Ecthyma (Exopol, San Mateo de Gállego, Spain), targeting the BL2 gene encoding a main viral envelope antigen. The kit contains a quantified synthetic positive control and endogenous control to avoid false-negative results. The amplification was performed in a FAST 7500 cycler (Applied Biosystems, Marsiling, Singapore) following manufacturer’s instructions.

### 2.7. Serological Analysis

Samples were also obtained by extracting 3 mL of blood from each lamb with a vacuum tube system without anticoagulant, providing serum for the detection of antibodies against ORFV. For the analysis, a homemade indirect ELISA was designed based on the recombinant immunodominant ORFV envelope protein 109 [19]. Recombinant 109 (109 rec) protein was synthesised, including the epitopes present in the first half of the protein. The dilutions that were used were made up of the 109 rec and other epitopes (1, 2, 3), with which various combinations were distinguished: 109+1, 109+1+2, 109+1+2+3.

Serum samples were taken prior to the experimental infection (Se0), pre-treatment (Se1, 8 days after inoculation) and post-treatment (Se2 at 13 days, Se3 at 20 days, Se4 at 27 days and Se5 at 34 days after the application of Tri-Solfen^®^, Sydney, Australia), with weekly periodicity. The Se5 sample was only taken from the 10 lambs that remained alive one month after infection, precisely to continue with their serological study.

Data were grouped according to the time the sample was taken (Se0, Se1, Se2, Se3, Se4, Se5) and according to the ELISA test (109+1, 109+1+2, 109+1+2+3). In addition, it was considered that the resulting values less than 0.3 were “negative” (absence of antibodies against contagious ecthyma), those greater than 0.3 were “positive” (presence of antibodies against ORFV) and those that were around 0.3 were categorised as “doubtful”.

### 2.8. Post-Mortem Study

One month after the experimental infection, 40 animals from both groups (20 of each) were subjected to humanitarian sacrifice with sodium pentobarbital. It was administered at a dose of 140 mg/kg, equivalent to 0.35 mL/kg, taking into account the weight of each lamb individually to perform the post-mortem study. A detailed necropsy was conducted, collecting the following information: number of lesions, anatomical location and severity, as previously explained. To evaluate and compare the severity between gross and microscopic lesions, tissue samples from ORFV-associated lesions obtained at necropsy were fixed in 10% neutral-buffered formalin, embedded in paraffin, and 4-mm thick sections were stained with hematoxylin and eosin (HE) and observed under a light microscope. The remaining ten lambs were sacrificed at the end of the experiment and followed exactly the same protocol as the previous ones. For the statistical study, all the animals were analysed together.

### 2.9. Statistical Analysis

All the data collected were integrated into a statistical matrix of the SPSS STATISTICS 26.0 program (IBM Corp., Chicago, IL, USA).

The comparison of proportions between groups (qualitative variables) was performed using Pearson’s Chi-square test and, alternatively, using Fisher’s exact test (some expected < 5). The Shapiro–Wilk test was used to establish the adequacy of the quantitative variables to the normal distribution. When the variables were not normal, non-parametric tests were applied for the comparisons between groups (non-parametric Mann–Whitney U test).

In normally distributed variables, one-way analysis of variance (ANOVA) was used to compare quantitative variables between groups and analysis of covariance (ANCOVA) was used when these comparisons could be affected by previous values of the variable considered.

The Kaplan–Meier method was used to determine whether the distribution of time (survival time) until the first lesion appearance from experimental infection differs between treatment groups. In this method, the event status shows two mutually exclusive and collectively exhaustive states: “censored” or “event” (where the “event” means “the first lesion is present” and “censored” means “the individual does not experience the event before the study ends”). The time to an event or censorship should be clearly defined and precisely measured. The time to an event is referred to as a complete data. Time to censorship is referred to as an incomplete time—until the moment of the last observation, which coincides with the end of the study. The animal is free of lesions, but it could continue to be free of lesions for a longer time. The time to event or censorship is always recorded and considered in the study for comparison between treatment groups. Therefore, the estimation of the mean survival time is limited to the largest (observed) survival time if it is censored. The comparison of survival times between groups was conducted using the Breslow test. In addition, the Bonferroni correction was applied in all cases of multiple comparisons.

In all the statistical tests with which the association between variables was determined, values of *p* < 0.05 were considered significant.

## 3. Results

### 3.1. Weighting

At time T0, upon arrival of the animals to the facilities, the mean total weight was 5.854 ± 1.1165 (SD) kg. The comparison between the means of the groups revealed no significant differences at the beginning of the experiment (5.958 ± 1.1777 (SD) kg in Box 1 vs. 5.758 ± 1.0711 (SD) kg in Box 2), being the two batches initially comparable in terms of weight.

Figure 1 shows the evolution of the average weight of the lambs per group during the entire experience. Significant differences were not found (*p* > 0.05) between the two groups.

### 3.2. Clinical Examination

Table 1 shows the days until the first lesion appearance from the experimental infection. The mean and SD of these times are displayed; the first vesicular lesions appeared on the right and left upper lip at a mean of 8.8 days and 8.6 days post-infection, respectively. The latest lesions appeared in the gums 24 days post-infection. Per box and globally, Table 1 also shows the total number of studied individuals (N total) and the number of individuals that did not experience lesions before the study ended (N censored). As noted above, both times to event or censorship are considered for comparison between treatment groups; therefore, N total means the number of animals included in each analysis. No significant differences (*p* > 0.05) were found between the groups in most locations. However, there were differences in the right lower lip (RLL) (*p* = 0.009) and in the “other locations” (*p* = 0.026), where the shortest average time of appearance was recorded in Box 2 (Table 1).

All 50 lambs ended the study with orf lesions, although the lesions appeared with a dispersion of 8 days minimum to 26 maximum. During the study period, no significant differences (*p* > 0.05) were found between the groups regarding lesion-free individuals, with high percentages of individuals with one or more lesions in both boxes. 

On the other hand, when evaluating the severity of the lesions, it was observed that there were significant (*p* < 0.05) and highly significant (*p* < 0.001) differences in some of the locations between the groups. In general, the most severe lesions were located on the right and left upper lips, while the least serious were those on the gums, which were the lesions that appeared later (24 days). The box 2 showed significantly higher severity lesions (“more severe lesions”) in the RLL, LLL and in other locations such as the nose or face (Table 2).

### 3.3. Haematological Analysis

Red series parameters, including erythrocytes, haematocrit, haemoglobin, mean corpuscular volume (MCV), mean corpuscular haemoglobin (MCH) and mean corpuscular haemoglobin concentration (MCHC), were found within the normal ranges in all the samplings in all the lambs analysed. In addition, no significant differences (*p* > 0.05) were detected between both batches at any of the sampling periods, with the only exception being erythrocytes measured in He0, where they were slightly below the lower limit in both groups (mean: 8.2250 ± 1.13620 (SD) M/μL in box 1 and 8.5908 ± 1.32259 (SD) M/μL in box 2). The reference value in sheep is considered to be 9.49. at 15.12 M/μL.

Analysis of the leukogram, including concentrations and percentages of total leukocytes, segmented neutrophils, lymphocytes and basophils, was within the reference values, and no significant differences (*p* > 0.05) were observed between the two batches in any of the parameters nor the measured times. Monocytes were above normal values (0.00–0.12 K/µL) in the two groups during He0 (0.45614 ± 0.36994 in Box 1 vs. 0.3559 ± 0.26951 in Box 2) and He1 (0.5900 ± 0.58874 in Box 1 vs 0.3419 ± 0.34816 in Box 2) (both prior to treatment with Tri-Solfen^®^), although there were no significant differences (*p* > 0.05) between them.

However, in the eosinophil count in concentration and percentage, despite being within the normal range, there were significant differences (*p* < 0.05) between Box 1 and Box 2 in He1 (pre-treatment) and the He3 (13 days post-treatment) (Table 3).

Finally, the mean platelet concentration and volume (MPV) were within the reference values and no significant differences (*p* > 0.05) were found between the means of the two batches.

### 3.4. Molecular Analysis 

The results obtained in the Real-Time PCR tests from the swabs of the lambs’ mouths are shown below (Table 4). 

In the first sampling (H0), prior to the experimental infection, no positive results were found in any of the batches, confirming the negativity of the lambs against the orf virus and thus providing the necessary condition to start the experiment. In H1, one-week post-treatment, there was an average of 73.3% negative animals in the two batches, while in Box 1 (control group) there was a total of 29.2% positives and in Box 2 (treated group), 23.8%. At this time, the percentage of positives was lower in the batch that had received treatment. However, seven days later (two weeks post-treatment), in H2, the trend changed, and the number of orf virus-positive lambs from Box 2 almost doubled that of Box 1, with 52.4% compared to 25.0%, respectively. Finally, in H3, 27 days after therapy, the results of the animals that remained after sacrifice (n = 10) were completely negative again in both batches. At none of the times in which qPCR was performed (H1, H2, H3), significant differences were detected between the batches.

### 3.5. Post-Mortem Study 

Gross and microscopic lesions coincided in severity (Figure 2), demonstrating that there were no differences between groups (Table 2). Additionally, histopathological analysis revealed that from grade 3, lesions commonly showed extensive ulcers with a diffuse severe neutrophilic infiltrate indicating bacterial contamination of ORFV-associated lesions.

## 4. Discussion

Contagious ecthyma has a worldwide distribution causing relevant economic losses in affected farms. In addition, it is a highly contagious disease. When severe outbreaks occur, all the lambs or kids on the farm can be affected, increasing the mortality rate and causing relevant weight loss. In a study on several commercial farms in England, with lambs naturally infected with contagious ecthyma, it was determined that there was a significant difference in weight when comparing affected and unaffected groups. Likewise, there was an important economic impact in the delay of the time to slaughter that this weight loss entailed [4]. As ORFV is very resistant in the environment, and ewes are asymptomatic carriers, it is considered impossible to eradicate from the affected farms, with control of this disease requiring vaccination. However, the vaccines that currently exist on the market are live attenuated virus vaccines and are not available in many European countries. Although these vaccines may confer short-lived immunity, and the attenuated strain of the vaccine virus can sometimes cause disease [20,21], the live orf vaccine is widely used in Australia and considered an important intervention on many sheep farms due to the effective suppression of the disease.

The treatment of the disease is based on the use of antiseptics and antibiotics to control secondary infections that are frequently produced by contamination of ecthyma lesions. With the significant reduction in the use of antibiotics imposed by the appearance of antimicrobial resistance, it is imperative to find a non-antibiotic treatment that controls both the spread of the virus and possible bacterial contamination of the lesions. For this reason, the present work was developed with the objective of testing the topical anaesthetic wound formulation TS developed in Australia for reduced pain and enhanced wound healing in surgical husbandry procedures. TS has also been shown to be efficacious for treating FMD in cattle and buffalo in Laos and Cameroon [14,15,16]. In both studies, treated cattle and buffalo achieved both superior appetite and lesion healing scores with a more rapid reduction in dimensions of lesions than other groups. Unfortunately, no significant differences in the present work have been found between the treated and untreated groups, nor the weights, clinical progression and gross and microscopic lesions. The foot-and-mouth disease virus (FMDV) causes vesicles by spreading in the horny cells of the skin and oral mucosa, comprising the most superficial layers of the epithelium [22]. However, the orf virus also produces modifications in deeper epithelial layers, such as the spinous and basal layers, even reaching the dermal cells [1,21], where the product applied topically could not act, probably explaining the contradictory results obtained in this study. 

Some significant differences were found in the haematological analysis in our study. Monocytes showed values above normal rates in the two groups during the two samplings prior to treatment with Tri-Solfen^®^, but there were no significant differences between groups. The chain of actions that occur in the non-specific immune response with the aim of minimising tissue injury begins with the activation of macrophages present in the injured tissue or through the release of a series of chemical mediators, such as cytokines, dispensed by monocytes and blood lymphocytes [23]. This could explain the monocytosis registered in the two batches of lambs after experimental infection. Further, although eosinophil values were within the normal range in both batches, significant differences were observed, with eosinophils being higher in the treated group. These results agree with the data obtained in the preliminary study on the use of Tri-Solfen^®^ in lambs naturally infected with the orf virus [24], where it was also found that the treated group had a significantly higher mean eosinophil count, although within the normal range. This could be due to a certain local hypersensitivity reaction that could be generated by the application of TS^®^ on the skin, which would justify the significant differences between both groups. However, the microscopic study did not show differences in the number of eosinophils in the dermis of both groups. Different moments were selected for the haematological, serological and molecular sampling based on studies conducted previously [24].

The previous studies speculated that TS could have a viricidal effect in FMD, limiting the transmission of the virus [14]. This could be related to the low pH of the product (2.7–2.9) that might lead to a reduction of viral activity, potentially limiting virus transmission during outbreaks. Likewise, the concentration of lidocaine in TS could also have a direct viricidal effect, as at concentrations ranging from 0.5 mg/mL (0.05%) to 100 mg/mL (10%), lidocaine blocks nociception and has been shown to exhibit antiviral activity against the herpesvirus [25]. In addition, we have the encouraging preliminary result obtained in the trial performed in 2021 [24], where, although no significant differences were found in the clinical progression of the lesions and PCR quantification between the cohorts, there was a significant difference in reduction in infective viral load obtained by culture between the groups, suggesting that treatment of early-stage lesions with TS may reduce the infective viral load present in orf lesions [24]. Nevertheless, in our current study, at none of the times in which qPCR was performed, significant differences were detected between the groups. Once again, these results could be explained by referring to the mode of action of the orf virus, capable of affecting the deepest layers of the epidermis as opposed to the penetration capacity of TS^®^, which could be insufficient to reach them and thus exert a possible viricidal effect, making the virus disappear.

The onset of seroconversion of the inoculated lambs occurred 35 days post-inoculation, although again, no differences between groups were observed. In the authors’ knowledge, this fact had not been previously studied. Therefore, this is the first report that shows that ELISA detects the first antibodies after infection with the orf virus from 35 days after infection. These results are consistent with the development of immunity through vaccination with live attenuated orf virus, which occurs from 4 to 8 weeks after receiving the vaccine [26].

Based on the results obtained, we can conclude that treatment with the tested topical antiseptic/analgesic has not offered efficacy against experimental infection with the orf virus and the development of contagious ecthyma lesions. However, an important aspect must be taken into consideration, and that is the early application of the product. The treatment was applied very early after the appearance of the lesions, at 8 and 11 days. Observing later that the presence of the lesions after the experimental infection can take up to 26 days, the last to appear being those of the gums. Thus, the lesions that appeared after day 11, which were the majority, did not receive TS treatment. Further, the treatment should be applied when the vesicles erupt to the outside, then, the product can come into contact with the virus and favour healing avoiding the pain caused by the lesions with the analgesic that TS carries. However, as we have observed, not all lesions emerge simultaneously; the dispersion is variable, making effective therapy a challenging task. In addition, the evolution of the orf lesions are vesicles, papules and pustules, with many of them becoming proliferative, further compromising the antiseptic and analgesic actions of the product in orf when compared to other more eruptive lesions as in FMD or necrotising lesions as in Lumpy skin disease [14,15]. 

## 5. Conclusions

Results of TS therapy in lambs experimentally infected with orf indicated that treatment had no effect on weight gain and clinical progression of the lesions. It was suggested that the deep basal epithelial location of the orf lesions may have prevented the therapy from having altered the clinical course. These results appeared less encouraging than those obtained in FMD, likely reflecting the relatively milder clinical signs of bovine FMD and the deep basal epithelial location of orf lesions. Nevertheless, further studies are necessary in natural infections and with a more significant number of applications of the product to be able to affirm if it improves the welfare of the animals and prevents the spread of the virus on the farm.

## Figures and Tables

**Figure 1 animals-13-02962-f001:**
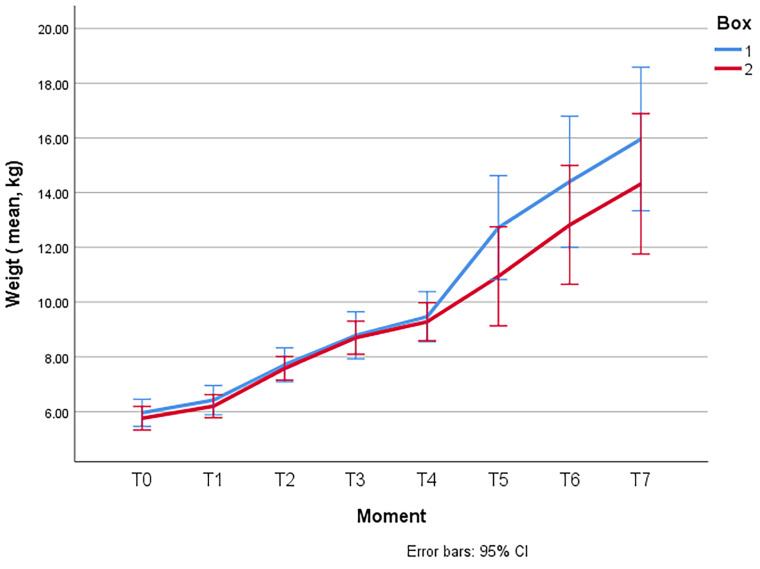
Evolution of the weights of the lambs of Box 1 (control) and Box 2 (treatment) throughout the entire study.

**Figure 2 animals-13-02962-f002:**
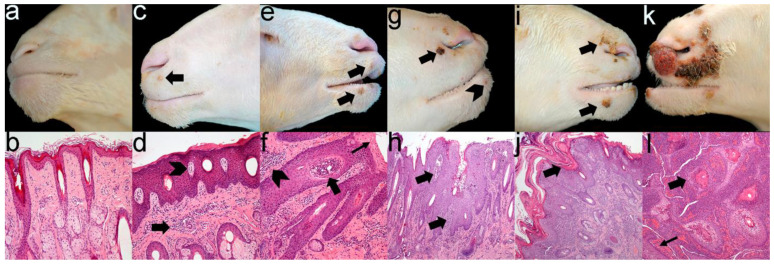
Severity grade of gross and microscopic ORFV-associated lesions. Grade 0 is classified as the absence of macroscopic (**a**) and microscopic (**b**) lesions. Grossly, grade 1 shows a low number of vesicles (arrow) or papules (**c**). Histologically, mild multifocal lymphoplasmacytic dermatitis (arrow) admixed with acanthosis and epidermal vesicles (arrowhead) are seen (**d**). Grade 2 is characterised by a higher number of papules and pustules (arrows) (**e**). Histology reveals moderate multifocal to coalescing dermatitis (arrowhead) admixed with marked acanthosis, parakeratotic hyperkeratosis (thin arrow) and epidermal pustules (arrow) (**f**). Grade 3 is composed of multifocal scab formations (arrow) with scattered pustules (arrowhead) (**g**), microscopically characterised by severe acanthotic proliferative dermatitis (arrows) (**h**). In grade 4, scabby proliferative dermatitis (arrows) starts to expand (**i**,**j**). Grade 5 shows multifocal to coalescence scabby proliferative and necrotising dermatitis (**k**), histologically observed as a total loss of skin architecture, replaced by lymphoplasmacytic and neutrophilic infiltrate (arrow) and haemorrhages (thin arrow) (**l**).

**Table 1 animals-13-02962-t001:** Days until the first lesion appearance from experimental infection a: Estimation of the mean is limited to the largest survival time if it is censored (the individual does not experience the event before the study ends). *p* values for comparison between treatment groups are also showed.

	Days the First Lesion Appeared after Experimental Infection					
RUL	Box	Mean	Standard deviation	N total	N censored	*p*
	1	7.52	1.323	25	2	
	2	10.144	1.087	25	3	
	Global	8.829	0.879	50	5	0.083
LUL	Box	Mean	Standard deviation	N total	N censored	*p*
	1	7.2	1.046	25	2	
	2	8.326	1.713	25	2	
	Global	8.638	1.354	50	4	0.920
RLL	Box	Mean	Standard deviation	N total	N censored	*p*
	1	14.167	1.113	23	8	
	2	11.287	2.244	27	3	
	Global	15.954	1.955	50	11	0.009
LLL	Box	Mean	Standard deviation	N total	N censored	*p*
	1	19.64	3.478	25	9	
	2	14.423	3.014	25	6	
	Global	17.155	2.341	50	15	0.496
Gums	Box	Mean	Standard deviation	N total	N censored	*p*
	1	26.52	3.245	25	13	
	2	23.125	3009	25	10	
	Global	24.857	2.23	50	23	0.850
Other locations	Box	Mean	Standard deviation	N total	N censored	*p*
	1	13.84	0.913	25	6	
	2	11.284	0.923	25	3	
	Global	12.585	0.675	50	9	0.026

**Table 2 animals-13-02962-t002:** Results of the mean gross severity of the lesions.

	Severity of the Lesions (1–5)								
RUL	Box	Mean	Standard dev.	*p*	LLL	Box	Mean	Standard dev.	*p*
	1	1.810	1.712			1	0.760	1.239	
	2	1.670	1.717			2	0.970	1.423	
	Global	1.740	1.715	0.203		Global	0.865	1.331	0.046
LUL	Box	Mean	Standard dev.	*p*	Gums	Box	Mean	Standard dev.	*p*
	1	1.760	1.600			1	0.630	1.168	
	2	1.850	1.685			2	0.660	1.253	
	Global	1.805	1.643	0.574		Global	0.645	1.211	0.973
RLL	Box	Mean	Standard dev.	*p*	Other loc.	Box	Mean	Standard dev.	*p*
	1	0.800	1.284			1	1.090	1.494	
	2	1.090	1.389			2	1.470	1.556	
	Global	0.945	1.337	<0.001		Global	1.280	1.525	<0.001

**Table 3 animals-13-02962-t003:** Results of the means of eosinophils of the lambs in He1 (15 days post-infection) and He3 (35 days post-infection) (*p* < 0.05).

		He1			
Eosinophils (%)	Box	Mean	Standard dev.	N	*p*
	1	0.9000	0.7314	21	
	2	2.0476	1.0930	21	
	Total	1.4738	1.0867	42	<0.001
Eosinophils (K/μL)	Box	Mean	Standard dev.	N	*p*
	1	0.0648	0.0530	21	
	2	0.1443	0.0822	21	
	Total	0.1045	0.0793	42	0.001
		**He3**			
Eosinophils (%)	Box	Mean	Standard dev.	N	*p*
	1	0.5050	0.4536	20	
	2	1.4182	1.8446	22	
	Total	0.9833	1.4322	42	0.008
Eosinophils (K/μL)	Box	Mean	Standard dev.	N	*p*
	1	0.0415	0.0341	20	
	2	0.1350	0.2298	22	
	Total	0.0905	0.1727	42	0.043

**Table 4 animals-13-02962-t004:** Table of qPCR results (*p* > 0.05). The subscript _a_ denotes that the column proportions do not differ significantly.

		Box		Total	*p*
		1	2		
		H0			
Negative PCR	Count/N	24/24	21/21	45/45	-
	% within Lot	100.00%	100.00%	100.00%	
Positive PCR	Count/N	0/24	0/24	0/45	
	% within Lot	0.00%	0.00%	0.00%	
		**H1**			
Negative PCR	Count/N	17/24 _a_	16/21 _a_	33/45	0.685
	% within Lot	70.80%	76.20%	73.30%	
Positive PCR	Count/N	7/24 _a_	5/21 _a_	12/45	
	% within Lot	29.20%	23.80%	26.70%	
		**H2**			
Negative PCR	Count/N	18/24 _a_	10/21 _a_	28/45	0.059
	% within Lot	75.00%	47.60%	62.20%	
Positive PCR	Count/N	6/24 _a_	11/21 _a_	17/45	
	% within Lot	25.00%	52.40%	37.80%	
		**H3**			
Negative PCR	Count/N	2/2	8/8	10/10	-
	% within Lot	100.00%	100.00%	100.00%	
Positive PCR	Count/N	0/2	0/8	0/10	
	% within Lot	0.00%	0.00%	0.00%	

## Data Availability

Not applicable.

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
