# Peer review of "Use of a Local Anaesthetic/Antiseptic Formulation for the Treatment of Lambs Experimentally Infected with Orf Virus"

_animals, 2023, doi:10.3390/ani13182962_

Round 1

Reviewer 1 Report

The work of Lacasta et al. aims to develop a new medication to relieve the pain and heal wounds caused by ORF virus in lambs.

The authors conclude that the use of this new product had no effect on the clinical course of the disease when comparing treated and control groups.

Similar results were obtained in an earlier work by the same authors in naturally ORF infected lambs, although a virus-killing effect of the drug was noted.

The work is carried out with great care and according to strict scientific protocols. Unfortunately, the results obtained have no useful implications for the treatment of infectious diseases causing skin lesions. In my opinion, the manuscript is overly broad in both the introduction and the materials and methods sections with a lot of sentences that are not necessary for understanding the work.

In my opinion, the manuscript should be published in the form of a short communication.

Author Response

Dear Reviewer, 

Thank you very much for your kind comments.

As you well know, the work that is developed for a research is independent of its results and the lack of expected results does not generally result in a bad research project.
The work carried out in this investigation was intense and conscientious, as the paper shows. We believe that summarizing this work in a short communication would reduce its credibility. In addition, it would be difficult to explain the results if you do not show the many studies carried out, which must be reflected in order to confirm the results correctly.
In the future new investigations will be carried out to confirm these results.

Thank you very much again and kind regards

Reviewer 2 Report

This manuscript presents some interesting but partial results about the effect of the treatment of viral contagious ecthyma (orf) lesions with a local anaesthetic/antiseptic wound formulation (Tri-Solfen, TS) in lambs experimentally infected with the virus. The overall effect of TS is less encouraging that the observed with other virus (such as FMDV) since there is no improvement in the clinical progression of the lesions, nor there is a positive effect on the weight gain of the lambs, nor on the amount of virus detected by qPCR in the control and treated groups of lambs. The authors argue that the lack of effect may be due to early application of treatment (probably most lesions did not receive adequate treatment), but also to the characteristics of the lesion itself (deep basal epithelial location). Thus, with data obtained in this study, the topical use of TS in cases of orf lesions is far from being a real option for the treatment of this viral infection, since no differences were observed between the control group (not treatment) and the group treated with TS.

There are several aspects that are not clear or need to be improved.

REFERENCES: Authors should carefully review the reference numbers (since there are two number 1, all the numbering is probably wrong). E.g., on page 13, line 450, reference 27 is cited (but the last number of the reference list is 26). All references included in the text must be reviewed and checked throughout the entire manuscript.

MATERIAL AND METHODS:

Line 162: Median tissue culture infectious dose must be TCID50. The authors should review and better explain the calculations of the viral titer per inoculum and the final dose received by each animal (lines 161-173)

It is not clear why samples for different types of analyses (haematological, qPCR, serological) are taken on different days during the experiment.

Line 208- 2.6. “Biomolecular analysis”: it would be better to call it directly “viral detection or viral quantification by PCR”. The days for sampling (H0, H1, H2 and H3) are different to those included in Table 4 (H0, H3, H4, H6), they need to be reviewed and correct. The commercial qPCR used is called RT-PCR in Table 4, lines 351, and 439.

Line 245. Post-morten study: What happens to the 10 lambs that are kept up to 35 days post infection? It is necessary to explain how they are sacrificed and if any post-mortem study is carried out.

RESULTS:

Figure 1: the description of box 1 (control) and box 2 (treatment) must be added.

Table 1: The title of the table is confusing ("survival analysis for the days…”), the superscript is not seen in the data. P value must be added to tittle (same as in table 2)

Severity of lesions (lines 302- 308 and Table 2): line 307- LID must be LUL. How many lambs are included in the study of the severity of lesions? No animals censored? (In Table 1, the authors included that “The data of those lambs that did not present lesions during the time that the study lasted or that died were censored”, so it is supposed that it would be the   situation in Table 2).

Table 3: In “N column” the total of animals should be included and a “N censored” column should be added with the animals excluded from the study (as in Table 1). Add the description of He1 and He 3 in the tittle of the table.

Table 4: “Count should be replaced by number (N), and again, a N censored data should be added. Add the description of H0, H3, H4, and H6 in the tittle of the table (after reviewing and checking with section 2.6). In H4, the names of "count" and "%within lot" are displaced. In H6, there should be only 10 animals (those that remained after the rest were sacrificed).

Lanes 341-352: The days for sampling (H0, H3, H4, H6) are different to those included in section 2.6 (H0, H1, H2 and H3) they need to be reviewed and correct.

Figure 1: it must be “Figure 2” (and change also in line 355). The letters g) and j) are missing at the figure legend. Gross and histological lesions are difficult to see, arrows should be included pointing out the most relevant in each image.

DISCUSSION:

Lines 433-438. In a previous trial on TS in experimentally orf-infected lambs, the authors did not found significant differences in PCR quantification between the cohorts, but they explain that “there was a significant difference in reduction in infective viral load between the groups, suggesting that treatment of early-stage lesions with TS may reduce the infective  viral load present in orf lesions [25]”. It difficult to understand this sentence as the viral load as quantified by qPCR.

Round 2

Reviewer 2 Report

I would like to thank the authors for following my recommendations and making many of the suggested changes.

However, I still have a few questions and doubts. 

The concept of "censored" is a bit confusing and should be better explained in the text (2.9 and results), not just in the Tables. It is difficult to understand the number of animals included in each analysis.

In addition, the discussion should also explain the reason for different sampling dates (haematological, PCR, serological) since they are based on a previous study.

Table 1: In the title of the table, write experimental infection instead EI

Severity of lesions (lines 302- 308 and Table 2): line 307- LID must be LUL. It has not been changed, and I can not find these data in this table.

It is not yet clear if any postmortem study is carried out on the 10 lambs slaughtered on day 35 post infection. And if it has been done, the results found.

Table 4: In H3, there should be only 10 animals (those that remained after the rest were sacrificed). It is not correct to include the data of all the animals in the study, they should be positive 0/10 and negative 10/10

Respect the self-citations, in my opinion are relevant and directly related with the study. The new wound formulation has been developed by the senior author, and has several previous studies on its efficacy in another viral livestock infection, and also experience in lamb ORF. Perhaps the least appropriate self-citation is number 12 (a review of sheep vaccines by the first author), and it could be replaced by a more recent one.

Round 3

Reviewer 2 Report

I would like to thank the authors again for reviewing and making the suggested changes. Thank you for adding the new text in point 2.9; now the concept of censored and the analysis with the Kaplan-Meir method are clear and well explained. Table 1 is also better understood with the text added in point 3.2.

However, I have a few more comments.

Different samples dates: Although it is understandable that it may take a long discussion to explain the reason for using different sampling dates, the authors should at least mention that it is based on a previous study (and add the reference).

Table 1: the authors must also change EI within the table.

Table 4 and H3 data: The H3 data is still incorrect in the Total column (why 16?). There should still be ten animals under study: 10/10 negative (not 26/16, that is impossible) and 0/10 positive (not 0/16). The authors should review and correct these data again.

Author Response

Please, see the attachement.
